# Cross-National Attunement to Popular Songs across Time and Place: A Sociology of Popular Music in the United States, Germany, Thailand, and Tanzania

**Tony Waters** [1,2,*] and **David Philhour** [3]

[1]  Department of Peace Studies, Payap University, Chiang Mai 50000, Thailand
[2]  Department of Sociology, California State University, Chico, CA 95929, USA
[3]  College of Behavioral and Social Sciences, California State University, Chico, CA 95929, USA; dphilhour@csuchico.edu
*   Correspondence: twaters@csuchico.edu

**Abstract:** This paper explores empirically Edward T. Hall's assertion about the role of musical elements, including rhythm recognition and what are called "ear worms" in popular culture. To test Hall's assertion, data were collected from the United States, Germany, Tanzania, and Thailand in 2015–2017 using a 26 brief "song intros." Data were also collected from exchange students from South Korea and Turkey. Survey responses were analyzed using factor analysis in order to identify patterns of recognition. It was found that there were indeed patterns of recognition apparently reflecting national boundaries for some song recognition, but others crossed boundaries. A separate analysis of patterned recognition comparing American youth under thirty, with elders over 60 indicated that there were also boundaries between age groups. Such experiments in music recognition are an effective methodology for Culture Studies given that musical elements are tied to issues of identity, culture, and even politics. Music recognition can be used to measure elements of such subconscious habitus.

**Keywords:** popular music; culture studies; Theodor Adorno; Edward T. Hall; attunement; youth culture

---

## 1. Introduction

In 1983, anthropologist Edward T. Hall asserted:

> It can now be said with assurance that individuals are dominated in their behavior by complex hierarchies of interlocking rhythms. Furthermore, these same interlocking rhythms are compatible to fundamental themes in a symphonic score, a keystone in the interpersonal processes between mates, co-workers, and organizations of all types on the interpersonal level within as well as across cultural boundaries. I am convinced that it will ultimately be proved that almost every facet of human behavior is involved in the rhythmic process. (Hall 1983, p. 153)

Hall's provocative statement emphasizes the functional nature of interlocking rhythms and musical themes for co-ordination of social action within cultural boundaries. The production and reception of such interlocking rhythms are presumably specific to people groups framed by such things as language, nationality, ethnicity, age, religion, etc., i.e., a range of social stratification categories.[1] Hall

---

[1]  Tagg (Tagg 1997, pp. 1–2) has a similar argument but is not quite as reductionist as E. T. Hall. In his paper about music and time, Tagg writes in his article "Understanding Musical Time Sense" that "Such varying sets of rules governing musical

is asserting that humans are engaged in a social "dance" with each other at intimate inter-personal levels, stretching across organizations like companies, clubs, associations, and even nations. This "dance" produces and recreates, he asserts, the very essence of "every facet of human behavior."

Indeed, following from Hall's logic, it can even be surmised that musical components are part of the boundaries between "us" and "them." Or to put it in Durkheim (1973) classical terms, it is elemental to the cohesive nature of the social bond. This is because familiarity with elements of music begin early in childhood, even before consciousness. Indeed, studies have shown that the child in the womb is "entrained" to the inter-locking rhythms of the mother both before and after birth (Hall 1983, p. 177), starting with her heartbeat and continue with the sounds that pass from the "soundscape" into the womb (see also Tagg 1997, p. 23). Such sounds reflect the social world in which the mother is embedded. Such attunement becomes more complex as the child is socialized into society (see e.g., Ball 2010, pp. 80, 121; Levitin 2006, pp. 222–25), and is "fixed" in the teenage years (i.e., about 14–20) which are a critical period for the establishment of musical tastes (Levitin 2006, pp. 225–26). It is widely understood that this attunement is rooted in psycho-physiological responses implanted culturally into neural networks (see e.g., Egermann et al. 2015).[2]

In other words, the rhythms and other musical elements Hall is writing about come from a created culture, at whatever age musical tastes are acquired. After all, except for perhaps the mother's heartbeat, attunement to sound is not innate but learned, emerging from the cultural context into which a child is born and socialized. Traditionally, such soundscapes were passed along aurally from ear-to-ear in the rhythms of song, poetry, verse, and the very sounds of everyday life (see Schafer 1993), including those of nature, like the sounds of crickets, or culturally produced rhythms like the pounding of a mortar and pestle, modern traffic, or even factories. Such rhythms indeed are often passed along in a way that they come to be seen as an element of the "natural world," as individuals are attuned to those interlocking rhythms which extend across great distances among people who share the transmitted rhythms, beats, and lyrics. A most impressive description of such oral transmission was identified in the late nineteenth century by W. E. B. DuBois, who wrote about the melodies, lyrics, and rhythms created in southern slave cabins of the United States that became nationally recognized "wild sweet melodies" of the "true American musician" spreading only from mouth to ear before recorded music was available in a semi-literate land (see (DuBois 1903, chp. 1). See also (Mauch et al. 2015) for a description of the evolution of popular music after 1960).

The classic sociologist Weber (1978) took up a similar theme when he described the relationship between early music production and capitalism in his essay about the history of the piano (see also Feher 1987). In a slightly different manner from Schafer or DuBois, Weber described a relationship between the rhythms of the symphony orchestra coordinated by the piano, and the industrial revolution. In particular, he described how the piano, an instrument organized around a 12-key scale, and a fixed Middle C was created as a product appropriate for the rationalized capitalist marketplace with its appeal to the emerging European middle classes.

Since the nineteenth century though, such rhythms, natural or otherwise, spread in a new way via a popular music created in a context of modern market capitalism, and which crosses cultural boundaries via radio airwaves and other electronic forms of reproduction. Music spread not just from voice-to-ear-to-voice in the manner that DuBois observed, but more importantly via the media of phonographs, radio, television, satellite, and other electronic means. As a result, rhythms, beats, lyrics, and the other elemental musical elements spread ever wider, and without the distortions and changes inherent to oral transmission. This new form of electronic transmission occurred in the form

---

structuration in different cultures and subcultures contribute strongly to the construction of ideology by establishing different symbolic universes of affective, gestural and corporeal attitudes or behavior." This statement is I think consistent with what Hall writes.

2　　The mechanism for understanding this is probably embedded in mirror neurons which are a basis for the empathy via which the culture of music, nature, speech, and all other linguistic elements are acquired. See (Waters 2014).

of commercial products created by music producers, and consumed by music consumers. Thus, unlike DuBois' day, the creation and consumption occurred in the context of capitalist markets where music is a commercial product obeying the rules of the capitalist marketplace (Adorno 1941).[3] Thus marketing experts use rationalized surveys, expensive production facilities, marketing, and "big data" (see e.g., BBC 2016) to evaluate both cultural production and reception of music.[4]

Theodor Adorno (1941) in the 1930s and 1940s began to describe the emergence of "popular music" in the context of the radio in particular (see also Storey 2006, pp. 64–67; Middleton 1990, pp. 34–62). Adorno critiqued this phenomenon, asserting that popular music is a rationalized product designed to elicit a common emotional response in people spread across a globalized world economy. The paradox of modern popular music he wrote, is that in using the power of music to elicit emotion, rationalized music eliminates individuality, and in the process creates an attuned worker suitable for the labor market, and presumably also a citizen suitable for the political sphere.[5]

Music producers created this synergistic relation between producer and consumer, Adorno wrote, by using music to elicit an emotional response. The melodies though are no longer "sweet and wild" as DuBois wrote about the orally transmitted "sorrow songs" of the American slave, but broken into a standardized, and predictable combinations suitable for efficient production in the rational marketplace. Adorno saw this new phenomenon "popular music" as a device to create the modern consumer needed by capitalism, while also eliminating the sharp edges of individuality which served so poorly the demands of capitalist markets for productive labor, and compliant consumers. The result Adorno (1941) wrote is a consumer product which conditions workers to march to the beat of the capitalist factory, while also weeping about the loss of individuality, to a point where the unity of the individual "begins to crack."

> When popular music is repeated to such a degree that it does not any longer appear to be a device but rather an inherent element of the natural world, resistance assumes a different aspect because the unity of individuality begins to crack. (Adorno 1941)

Adorno goes on to emphasize that mass-produced popular music creates a world in which consumers

> Consume music in order to be allowed to weep. They are taken in by the musical expression of frustration rather than by that of happiness. The so-called releasing element of music is simply the opportunity to feel something. But the actual content of this emotion can only be frustration. Emotional music has become the image of the mother who says, "Come and weep, my child." It is catharsis for the masses, but catharsis which keeps them all the more firmly in line. One who weeps does not resist any more than one who marches. Music that permits its listeners the confession of their unhappiness reconciles them, by means of this "release", to their social dependence.

The formulaic "The Wall" by Pink Floyd, in which the band sings about the frustration of modern life as "just another brick in the wall" is perhaps the most ironic illustration of the type of formulaic music Adorno is writing about, at least in the English-speaking world. Axis of Awesome's "Four

---

[3] For a critique of Adorno's pessimism, and appreciation of Adorno's ideas about commodification of music, see (Frith 1998, pp. 13–14).

[4] Cultural theorists of the Birmingham School like Stuart Hall and Raymond Williams (see Storey 2006), have developed this point broadly, particularly in the context of modern capitalism.

[5] The attuned worker is in Max Weber's term a "disciplined worker" whose psycho-biological being is rationalized to the factory, bureaucracy, or other large rationalized social institution. Weber writes: "the point of such rational discipline is that rationalized orders are executed when received in a predictable fashion. This happens because the execution of any received command emerges from tactical responses that are conditioned reactions to precise drills. In the context of such drills, all personal critique is unconditionally deferred, and personal convictions are constantly adjusted towards the pre-determined goal reflected in how the received order is executed." Predictability, tactical responses, and precise drills are all undertaken reference to underlying culturally-generated rhythm. See (Weber 2015, p. 59; Waters 2018).

Chord Song" is another presumably profitable, but certainly satirical critique of the dominant chord progression I, V, VI, IV which seemingly delights English-speaking audiences in Europe, Australia, and North America. Both songs satirize in different ways the ubiquity of the "inter-locking rhythms" Hall wrote about.

The popular music that Adorno critiques spread around the world only since the 1930s or so with the mass production of radios and phonographs. However, despite much popular music becoming globally recognized, what is consumed still maintains variations across both time and place. For example, Adorno wrote about popular musicians like Count Basie, Frank Sinatra, and others who were popular in the 1930s and 1940s. These were the musicians that stimulated the longings—the weeping and marching—of a specific generation in the United States, and perhaps elsewhere, tying a generation into a common culture of consumption. In succeeding generations, other musicians like The Beatles of Liverpool, Pink Floyd of London, Michael Jackson of Los Angeles, and Psy of Seoul did the same thing, taking advantage of the mass appeal of interlocking rhythms.[6] But is Adorno's all-inclusive reductionist assumption about a global mono-culture really as bland as he suggests? Do other patterns persist which are specific to national cultures, particular age cohorts, "within as well as across cultural boundaries" as Hall asks?

This is a question that the first author asked while teaching American and German undergraduates between 1996 and 2014. The longer he taught, the less relevant were the songs of his own youth in the 1970s and 1980s. In particular, students who in the late 1990s recognized the folk songs and television introductions of his youth, no longer did so after about 2005.

With the advent of YouTube with its widely available music clips, we began to evaluate what patterns different audiences would recognize or not, with a focus on variation in both generation within the United States, and cross-nationally between the United States and Germany. With the advice and assistance students, we developed a play list of song intros (also known colloquially as "ear worms") which would be sensitive to the musical experiences of different ethnic groups found in our Chico classrooms, and Germany, Thailand, and Tanzania where we would be traveling in 2015–2017.[7] The 26 song introductions were then tested among convenience samples which included students at CSU Chico of American and Korean undergraduates, elders in a "current events" at a Lutheran Church in nearby Grass Valley, California, students at Midland University in Nebraska, USA, classrooms at Leuphana University in Germany, Stefano Moshi University College in Tanzania, and Payap University in Thailand. The surveys were undertaken in 2015–2017.

## 2. Methods

### 2.1. Selection of the Song Intros

The song intros were selected after informal experimentation in Germany, and especially in California. In California, students in classes were consulted about what they thought should be included in a sample selection. It soon became obvious that there would be variation across ethnic groups in the California classroom, particularly Chicano/Latino, and African-Americans. It was also apparent that song intros which spanned generations were needed.

Other song intros were listed below were the context of the senior author's travel plans in 2015–2017. This is why intros from Germany, Thailand, and Tanzania were included. Song intros from popular songs from Ukraine and China were included as controls.

---

6    Cultural sociologists of popular music like Frith (1998, 2004), Middleton (1990) developed this point about the relationship between music and capitalists marets further.
7    Marcos Zepeda contributed to the selection of songs, and testing them with his fellow students.

## *2.2. Description of the Intros*

The following song Intros were selected for the survey, and compiled into a four-minute long set of 26 song intros, which was uploaded to Google Drive. Each intro was only 4–8 s, and the total length of the recordings was minutes.

## *2.3. Survey Instrument*

The respondents were gathered in groups, and asked whether they recognized a song intro or not, and instructed to answer yes or no. Students were also asked to indicate any association they had with a particular song intro (Box 1).

**Box 1.** Track names (d).

1 Let It Be—The Beatles. An iconic 1970 hymn of popular commercial music. From England.
2 Free Bird—Lynyrd Skynyrd. A popular song from the 1974 in the United States.
3 South Africa/Tanzanian National Anthem. Played frequently at the 2010 World Cup Games.
In South Africa, and the melody (with different lyrics) is recognizable in Tanzania.
4 America the Beautiful—version which was part of a Coca-Cola commercial in 2014.
5 Don't Stop—Fleetwood Mac—a 1977 song popular in the 1980s and 1990s.
6 Made in Thailand. By the pop group Carabao. This is a popular Thai protest song from 1984
7 Ode To Joy. Beethoven, vocal version.
8 Somewhere- Judy Garland, 1938 Wizard of Oz.
9 60 Minute Clock, Long running US news program which started in 1968.
10 The Moon Represents my Heart. Popular Chinese song by Theresa Tang from 1977.
11 Chervona Ruta Popular Ukrainian Song form 1968.
12 Somewhere—by Israel K. in 1990. Ukulele version of the Wizard of Oz song
13 I Love Lucy Theme. Popular US TV show in 1950s-1970s, original broadcast in 1951–1957.
14 Happy—Pharrell Williams. Popular song released in 2013.
15 Für Elise. More Beethoven. Used in piano lessons, movies, and ring tones in the early 2000s.
16 TID—Zeze Popular Tanzanian Song from the early 2000s.
17 Fresh Prince of Bel Air Theme. Popular US television show broadcast from 1990–1996.
18 Los Tigres del Norte, "La Mesa del Rincon" Popular north Mexican song in the last part of twentieth century.
19 Ice Ice Baby—Vanilla Ice. Popular in the 1990s and 2000s.
20 As Time Goes By, from classic movie Casablanca from the 1940s
21 We Shall Overcome. US Civil Rights anthem of the 1960s popularized by Martin Luther King
22 Born To Run by Bruce Springsteen. Popular song in late 1970s
23 Tagesschau (Germany). Intro to German news program, every day at 8 p.m.
24 Loi Katong (Thailand). Popular traditional holiday song in Thailand. This version uses a classic Thai pentatonic scale
25 Marlboro cigarette commercial from before 1969.
26 Teach the World to Sing. 1971 Coca-Cola commercial in the early 1970s.

## *2.4. Administration of the Survey*

The survey was administered to 358 respondents. The self-reported nationalities included 158 people form the United States, 34 Thai students, 33 Tanzanian students (all male), 10 students from Turkey, 36 Germans, 34 from South Korea, and 62 from other countries. There were 26 nationalities, including a small number who indicated they were dual nationals/identities. In total, 119 of the respondents were male, 212 female, and 2 were an "other" gender. 25 had missing data for gender.

(1)　Sociology classrooms in Chico, California, 2015 (N = 70).
(2)　Korean Exchange students in Chico, California, 2015 (N = 31).
(3)　German and international students at Leuphana University, Germany 2015 and 2016 (N = 57).
(4)　International and Thai students at Payap University, Chiangmai, Thailand 2016, (N = 92).
(5)　Undergraduate students at Stefano Moshi Memorial University (SMUCCO), Moshi, Tanzania 2016 (N = 33).
(6)　Undergraduate Students at Midland University in Midland Nebraska 2015, (N = 33).

(7)　　Grass Valley, California, Church, Adult Education Class (N = 40). 2015.

At least thirty respondents were collected from the following countries: The United States (three groups), Thailand, Germany, and Korea. At least five responses were collected from students from Turkey, Myanmar, China, and Japan.

Notably these convenience samples do not "represent" national groupings. Rather they are a snapshot of a particular time, place and population which is interpretable to the other populations for whom similar data were collected. As a short-hand though, we refer to them by nationality to facilitate interpretation relative to the other samples collected for this study. Nationality in this sense is assumed to be a reflection of where the responses were socialized as children and first exposed to musical elements.

## 3. Results

Quantitative and Qualitative results are summarized in the following tables. These results indicate indeed that there are patterns of "interlocking rhythms" which tie some song intros to each other, as well as patterns of nationality and age. The quantitative and qualitative results are summarized below.

### *3.1. Results (Quantitative)*

Results were analyzed using a combination of factor analysis, correlations statistics between songs (negative and positive), and a comparison of the recognition patterns of younger and older Americans (*t*-test). As described below, the results consistently indicated correlations between the recognition of some songs, and not others. Recognition reflected a mix of musical tastes, national identity, and age demographics.

3.1.1. Factor Analysis.

Figure 1 reflects the best-fit scree plot used for the factor analysis. A four-factor analysis was chosen because it accounted for 45.6% of the variance, and as described below was readily interpretable. (A five-factor solution proved uninterpretable) (See Table 1).

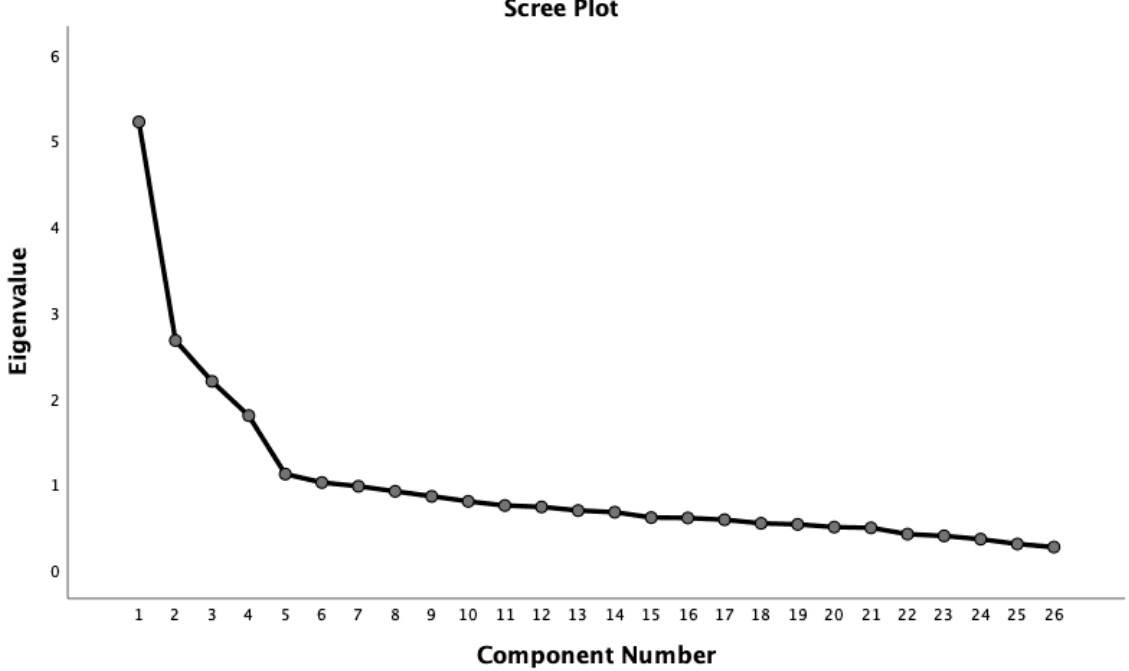

**Figure 1.** Scree Plot.

**Table 1.** Analysis 4 component solution.

| | | | | |
|---|---|---|---|---|
| **Rotated Component Matrix** [a] | | | | |
| **Song** | **Component** | | | |
| | **1** | **2** | **3** | **4** |
| Ice Ice Baby | 0.763 | | | |
| Fresh Prince of Bel Air | 0.704 | | | |
| Born to Run | 0.632 | | | |
| Free Bird | 0.567 | | | |
| 60 Minutes Clock | 0.531 | | | |
| Somewhere—Iz | 0.528 | | 0.333 | |
| Happy | 0.523 | −0.495 | | |
| Don't Stop | 0.506 | | | |
| Let it Be | 0.417 | | 0.334 | |
| Teach the World to Sing | | 0.727 | | |
| As Time Goes By | | 0.673 | | |
| We Shall Overcome | | 0.672 | | |
| America the Beautiful | 0.525 | 0.561 | | |
| Marlboro Commercial | 0.378 | 0.527 | | |
| Los Tigres del Norte | | 0.488 | | |
| I Love Lucy | | 0.416 | | |
| TID—Zeze | | | −0.667 | |
| Fur Elise | | | 0.649 | |
| Somewhere | 0.309 | | 0.644 | |
| Ode to Joy | | | 0.642 | |
| So. African National Anthem | | 0.326 | −0.549 | |
| Moon Represents | | | | 0.682 |
| Loi Katong | | | | 0.670 |
| Chevrona Ruta | | | | 0.583 |
| Made in Thailand | | | | 0.567 |
| Tagesschau | | | | 0.449 |
| Extraction Method: Principal Component Analysis. | | | | |
| Rotation Method: Varimax with Kaiser Normalization. [a] | | | | |

[a] Rotation converged in 6 iterations.

*Factor 1* (nine songs). This factor reflects the globalization of some songs but not others. These were songs for which the Americans and Germans showed similar patterns of recognition.

*Factor 2* (seven songs). These reflect American standards which come out of American culture.

*Factor 3* (five songs). The Tanzanian data on this factor are either very strong relative to other countries, or extraordinarily weak relative to the three groups recognized elsewhere (see also Figure 2). Ironically, this is also where the classical Beethoven songs appear, which were recognized in all groups *except* Tanzania. Tanzanians did not recognize the Beethoven, but of course recognized the two Tanzanian/South African songs. This resulted in the peculiar loading of the most popular songs in every country except Tanzania (i.e., Beethoven), because it had strong *negative* values, and the songs which only were popular in Tanzania on the same factor. Note though that this loading reflects opposite patterns (opposites attract on this factor!).

*Factor 4* (five songs). This is the Thailand/Turkey cluster (four songs), though the German "Tagesschau" also appears here, as does the "control," the Ukrainian song Chervona Ruta. This factor is close to being uninterpretable by itself, though when it is evaluated in the context of Figure 2, there is seen some patterning between Thailand and Turkey.

It does turn out that the factor analysis is effective in identifying populations which share underlying familiarity with melodic and rhythmic elements. In this case, it is clear that the songs selected using Principle Component Factor Analysis as a data reduction technique is effective. This becomes clearest when the analysis is broken into the six countries where the respondents were raised, and is why such a small number of factors explained 45.6% of the variance (see Figure 2). Most notably in Figure 2,

it becomes clear that Factor 4, which is seemingly uninterpretable at first glance, reflects shared recognition patterns between Thailand and Turkey, despite the small number of students from Turkey surveyed.

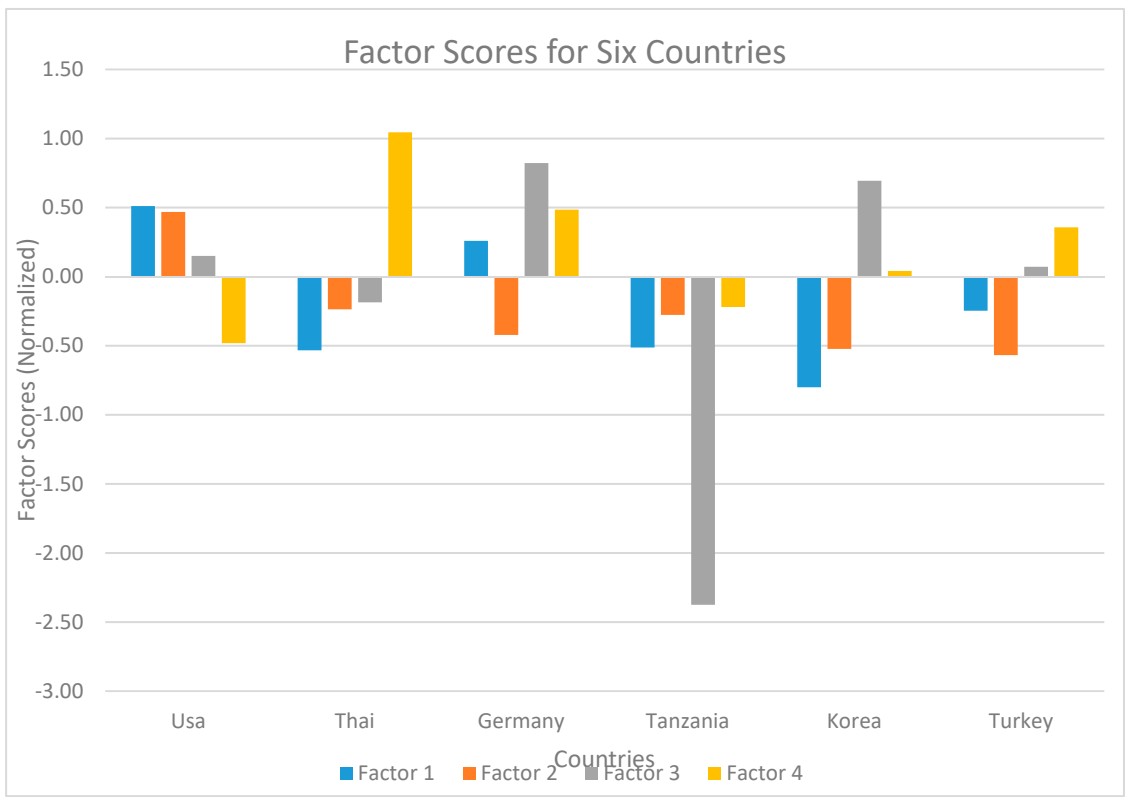

**Figure 2.** Component Factor Analysis by Country (4 × 6).

### 3.1.2. Positive Correlations between Songs

Tables 2 and 3 summarize the findings in a 26 × 26 table of correlation coefficients. Table 2 is in some respects redundant to the Factor analysis in that similar clustering is observable, albeit with less precision. In evaluating Table 3, a number of things stand out, particularly the fact that the two "Tanzanian" songs (TID Zeze and South African National Anthem) have strong *negative* correlations with the other songs, including those from the west and Asia.

**Table 2.** Correlations between each of the 26 song intros (26 × 26 matrix). Correlations that were above 0.5 (one pair) are indicated in bolded 16 point type. Correlations between 0.400 and 0.499 are indicated in bolded 12 point type. Correlations between 0.300 and 0.399 are indicated in standard 10 point type.

| Song Title | Correlates With | Researchers Comments |
| --- | --- | --- |
| (1) Let it Be | Free Bird | The Beatles Are Less Enduring than we Thought |
| | Somewhere | |
| | Fuer Elise | |
| | Ice Ice Baby | |
| (2) Free Bird | Let it Be | Very American set of correlations |
| | America the Beautiful | |
| | Don't Stop | |
| | Somewhere—Iz Version | |
| | Fresh Prince | |
| | Ice Ice Baby | |

**Table 2.** *Cont*.

| Song Title | Correlates With | Researchers Comments |
|---|---|---|
| | Born to Run | |
| (3) South African National Anthem | TID Zeze | Tanzania |
| (4) America the Beautiful | **Fresh Prince** | Most American set |
| | **Marlboro** | |
| | Free Bird | |
| | Somewhere | |
| | I Love Lucy | |
| | Los Tigres Del Norte/La Mesa del Rincon | |
| | We Shall Overcome | |
| | Born to Run | |
| | I'd Like to Teach the World to Sing | |
| (5) Don't Stop | **Born to Run** | American baby boomers |
| | Free Bird | |
| | Somewhere—Iz | |
| (6) Made in Thailand | Moon Represents | Chinese music made it to Thailand, but not to Germany, California, or Tanzania |
| (7) Ode to Joy | **Somewhere** | The correlations with the German Tagesschau is surprisingly low |
| | Fuer Elise | Two songs from Beethoven correlate, though not as strongly as others |
| (8) Somewhere | **Ode to Joy** | |
| | Let it Be | |
| | America the Beautiful | |
| | Somewhere—Iz | |
| | Ice Ice Baby | |
| (9) 60 Minutes Clock | Fresh Prince | Gen X recognizes the 60 Minutes clock, but not older people |
| | Ice Ice Baby | |
| (10) Moon Represents | Made in Thailand | China and Thailand go together |
| | Loi Katong | |
| (11) Chervona Ruta | (None) | This was included as a control. |
| (12) Somewhere—Iz | Free Bird | |
| | Don't Stop | |
| | Somewhere | |
| | Fresh Prince | |
| (13) I Love Lucy | Don't Stop | Older Americans who remember the 1992 presidential campaign? |
| (14) Happy | **Ice Ice Baby** | Only one strong correlation. Probably due to the more global recognition |
| (15) Fuer Elise | Let it Be | This goes across generations, and from Germany to the US |

**Table 2.** *Cont*.

| Song Title | Correlates With | Researchers Comments |
|---|---|---|
| | Ode to Joy | |
| | Ice Ice Baby | |
| (16) TID Zeze | South African National Anthem | Tanzania |
| (17) Fresh Prince | **Ice Ice Baby** | Younger Americans are patriotic Americans, too. |
| | **America the Beautiful** | |
| | Free Bird | |
| | 60 Minutes | |
| | Somewhere—Iz | |
| (18) Los Tigres del Norte | America the Beautiful | Perhaps Latinos are patriotic—and elderly like Latino music, too. |
| | As Times Go By | |
| | We Shall Overcome | |
| (19) Ice Ice Baby | **Fresh Prince of Bel Air** | Gen X American youth anthems |
| | **Happy** | |
| | Let it Be | |
| | Free Bird | |
| | Somewhere | |
| | 60 Minutes | |
| | Born to Run | |
| (20) As Time Goes By | **We Shall Overcome** | Elderly and Latinos |
| | **I'd Like to Teach** | |
| | Lost Tigres del Norte | |
| (21) We Shall Overcome | America the Beautiful | American Patriotism from the elderly and Latinos |
| | Los Tigres del Norte | |
| | Marlboro | |
| | I'd Like to Teach the World to Sing | |
| (22) Born to Run | Don't Stop | |
| | Free Bird | |
| | America the Beautiful | |
| | Ice Ice Baby | |
| (23) Tagesschau | (None) | |
| (24) Loi Kathong | Moon Represents | |
| (25) Marlboro | **America the Beautiful** | Marlboro was generally equated with "Bonanza" |
| | I'd Like to Teach the World to Sing | |
| (26) I'd Like to Teach the World to Sing | **As Time Goes By** | Coca-cola, Casablanca, Civil Rights, and Cigarettes. |
| | **We Shall Overcome** | |
| | America the Beautiful | |
| | Marlboro | |

**Table 3.** Negative Correlations. This Table includes all the statistically significant negative correlations (two asterisks indicate significance at the 0.01 level, and one at the 0.05 level). Most of the significant negative correlations were with the two Tanzanian songs, though there were also three negative correlations with the 2014 song "Happy" and three of the older American songs. The only songs for which there was a weak positive correlation with the South African National Anthem were "I'd Like to Teach the World to Sing," and "As Times go By" (statistics not shown).

| Song Title | Correlates With | Correlation | Researchers Comments |
|---|---|---|---|
| South African National Anthem | Let It Be | **−0.107 *** | Tanzania Missed on Beethoven and the Beatles |
| | Ode to Joy | −1.12 * | |
| | Somewhere | −0.174 ** | |
| | Fuer Elise | −0.198 ** | |
| | Ice Ice Baby | −0.124 * | |
| Happy | As Time Goes By | −0.190 ** | Elders missed 2014 pop hit Happy |
| | We Shall Overcome | −0.208 ** | |
| | I'd Like to Teach the World to Sing | −0.231 ** | |
| Tagesschau | I'd Like to Teach the World to Sing | −0.114 * | |
| TID Zeze | Fuer Elise | −0.339 ** | Tanzanians again—the missed out on Beethoven and the Beatles |
| | Somewhere | −0.276 ** | |
| | Ode to Joy | −0.268 ** | |
| | America the Beautiful | −0.155 ** | |
| | Let it Be | −0.120 * | |

Song Intros focused by strong nationalistic sources, e.g., American the Beautiful, the German news introduction, and the Thai song "Made in Thailand" do not have correlations with each other, and are country specific for recognition.

A number of more general songs, like "Let it Be," "The Moon Represents My Heart," "Ice, Ice, Baby" show strong correlations with songs, except those from Tanzania. The two Tanzanian songs show a strong correlation with each other, but fairly consistent negative correlations like "Let It Be," "Für Elise," or the Thai and Chinese songs.

There are also apparently strong relationships between songs which have a generational component for Americans. For example "As Times Go By," "We Shall Overcome," and "I'd Like to Teach the World to Sing" all have strong positive correlations with each other.

The Asian songs show strong correlations with each other, too (e.g., "The Moon Represents My Heart," "Made in Thailand," etc.), but do not have the negative correlations with the western songs like "Let it Be," "Free Bird," etc. Rather, the correlations are typically weakly positive and not at statistically significant.

3.1.3. Negative Correlations between Songs

Table 3 summarizes the anomalous negative correlations, which are also reflected in the data from the factor analysis for Factor 3 (see Table 1). As in the factor analysis, it is the Tanzanians that are the outlier, particularly with respect to classical western music. Such western songs though are recognized by Korean and Thai students.

## 3.2. Country Breakdowns

The five countries (USA, Germany, Thailand, Tanzania and Korea) with at least 30 cases, plus Turkey (10 cases) were separated out of separate analysis (see Table 4). Chi Square and Eta (ANOVA) values highlight the centrality of "America the Beautiful," "Somewhere," "Für Elise," and "TID Zeze" to this collection (Table 5). Particularly in Table 4, again the unique nature of the Tanzanian recognition patterns was apparent. It is in Table 4 that it is evident that only "Happy" by Pharrell Williams is recognized across all six countries.

**Table 4.** By nationality.

| Song Clip | Percent Recognizing Particular Song Clips | | | | | |
| --- | --- | --- | --- | --- | --- | --- |
| | USA | Turkey | Thai | Tanzania | Korea | Germany |
| | (N = 149) | (N = 10) | (N = 34) | (N = 33) | (N = 34) | (N = 36) |
| Ice Ice Baby | 72% | 56% | 30% | 13% | 29% | 89% |
| Fresh Prince of Bel Air | 56% | 22% | 6% | 13% | 3% | 44% |
| Born to Run | 38% | 22% | 15% | 9% | 3% | 31% |
| Free Bird | 57% | 22% | 21% | 16% | 16% | 47% |
| 60 Minutes Clock | 52% | 33% | 21% | 9% | 19% | 25% |
| Somewhere - Iz | 79% | 78% | 64% | 16% | 42% | 97% |
| Happy | 72% | 100% | 88% | 75% | 61% | 94% |
| Don't Stop | 42% | 0% | 30% | 16% | 19% | 36% |
| Let it Be | 65% | 33% | 42% | 13% | 84% | 83% |
| America the Beautiful | 92% | 11% | 18% | 16% | 6% | 36% |
| Marlboro Commercial | 79% | 56% | 36% | 19% | 13% | 56% |
| I Love Lucy | 66% | 44% | 39% | 34% | 32% | 31% |
| Los Tiegres del Norte | 43% | 22% | 21% | 13% | 3% | 28% |
| Teach the World to Sing | 46% | 11% | 30% | 22% | 6% | 22% |
| As Time Goes By | 35% | 22% | 36% | 13% | 29% | 25% |
| We Shall Overcome | 55% | 33% | 33% | 6% | 16% | 56% |
| Somewhere | 97% | 33% | 58% | 13% | 97% | 94% |
| Fur Elise | 97% | 100% | 88% | 25% | 94% | 100% |
| TID—Zeze | 3% | 0% | 15% | 75% | 6% | 6% |
| So. African National Anthem | 19% | 22% | 9% | 72% | 3% | 14% |
| Ode to Joy | 72% | 78% | 55% | 6% | 77% | 97% |
| Tagesschau | 14% | 44% | 27% | 3% | 42% | 100% |
| Made in Thailand | 11% | 11% | 76% | 13% | 3% | 17% |
| Chevrona Ruta | 7% | 33% | 27% | 13% | 6% | 11% |
| Moon Represents | 23% | 11% | 64% | 19% | 35% | 36% |
| Loi Katong | 9% | 22% | 36% | 13% | 13% | 14% |

**Table 5.** Using the Song as the Dependent Variable, Chi-square was calculated across the song Intro, and the data from six nations (United States N = 149), Thailand (N = 34), Germany (N = 36), Tanzania (N = 33), Korea (N = 33), Turkey (N = 10). (Total N = 295). Reported are the Eta values (ANOVA), and Chi-Square statistic. All were significant at the 0.01 level or less, except Don't Stop, and Chevrona Ruta.

| Song | Eta | Chi-Square |
| --- | --- | --- |
| (1) Let it Be | 0.424 | 53.088 |
| (2) Free Bird | 0.371 | 40.659 |
| (3) South African National Anthem | 0.464 | 63.843 |
| (4) America the Beautiful | 0.755 | 168.289 |
| (5) Don't Stop | 0.242 | 17.338 * |
| (6) Made in Thailand | 0.513 | 78.030 |
| (7) Ode to Joy | 0.501 | 74.224 |
| (8) Somewhere | 0.734 | 159.446 |
| (9) 60 Minutes Clock | 0.338 | 33.800 |
| (10) Moon Represents | 0.282 | 22.047 |

**Table 5.** *Cont.*

| Song | Eta | Chi-Square |
|---|---|---|
| (11) Chevrona Ruta | 0.228 | 15.381 * |
| (12) Somewhere—Iz | 0.449 | 73.784 |
| (13) I Love Lucy | 0.230 | 58.77 |
| (14) Happy | 0.259 | 19.796 |
| (15) Fuer Elise | 0.709 | 148.631 |
| (16) TID Zeze | 0.654 | 126.294 |
| (17) Fresh Prince | 0.444 | 58.178 |
| (18) Los Tigres del Norte | 0.305 | 27.367 |
| (19) Ice Ice Baby | 0.502 | 74.119 |
| (20) As Time Goes By | 0.172 | 8.727 ** |
| (21) We Shall Overcome | 0.376 | 41.674 |
| (22) Born to Run | 0.293 | 25.302 |
| (23) Tagesschau | 0.637 | 119.783 |
| (24) Loi Kathong | 0.239 | 16.781 * |
| (25) Marlboro | 0.466 | 74.442 |
| (26) I'd Like to Teach the World to Sing | 0.298 | 26.050 |

Chi-square Likelihood 0.001–0.10 = * (three songs) ** = 0.12 (one song).

### 3.3. Age Breakdown—Americans over 60 and under 30

Based on what was observed in the correlations coefficients, Table 6 was created to compare the American respondents into two categories based on age, i.e., those under 30 years old (N = 95), and those over 60 years old (N = 38). This was done to highlight the strong influence of age on recognition patterns. In total, 17/26 song intros had statistically significant differences in recognition, despite the small number of elderly sampled. Where the t-test was a more than a 0.05 probability and therefore not statistically significant, it is indicated with "ns".

**Table 6.** For Americans by young and old age cohorts (N = 133).

| | 17–30 Years Old (N = 95) | 60 and Older (N = 38) | Difference | Independent $t$-Test |
|---|---|---|---|---|
| (1) Let it Be | 0.69 | 0.55 | 0.14 | 1.50 ns |
| (2) Free Bird | 0.59 | 0.42 | 0.17 | 1.76 ns |
| (3) South African National Anthem | 0.15 | 0.26 | 0.11 | −1.43 ns |
| (4) America the Beautiful | 0.87 | 1.00 | 0.13 | −3.69 $p < 0.001$ |
| (5) Don't Stop | 0.42 | 0.26 | 0.16 | 1.78 ns |
| (6) Made in Thailand | 0.09 | 0.11 | 0.02 | −0.18 ns |
| (7) Ode to Joy | 0.61 | 0.95 | 0.34 | −5.41 $p < 0.001$ |
| (8) Somewhere | 0.96 | 1.00 | 0.04 | −2.03 $p < 0.05$ |
| (9) 60 Minutes Clock | 0.62 | 0.21 | 0.41 | 4.91 $p < 0.001$ |
| (10) Moon Represents | 0.20 | 0.23 | 0.03 | −0.47 ns |
| (11) Chervona Ruta | 0.06 | 0.05 | 0.01 | 0.23 ns |
| (12) Somewhere—Israel K. | 0.84 | 0.61 | 0.23 | 2.67 $p < 0.01$ |
| (13) I Love Lucy | 0.56 | 0.79 | 0.23 | −2.74 $p < 0.01$ |
| (14) Happy | 0.95 | 0.08 | 0.87 | 18.91 $p < 0.001$ |
| (15) Fuer Elise | 0.96 | 1.00 | 0.04 | −2.03 $p < 0.05$ |
| (16) TID Zeze | 0.03 | 0.05 | 0.02 | −0.57 ns |
| (17) Fresh Prince | 0.79 | 0.03 | 0.76 | 15.38 $p < 0.001$ |
| (18) Los Tigres del Norte | 0.36 | 0.61 | 0.25 | −2.65 $p < 0.01$ |

**Table 6.** *Cont*.

| | 17–30 Years Old (N = 95) | 60 and Older (N = 38) | Difference | Independent *t*-Test |
|---|---|---|---|---|
| (19) Ice Ice Baby | 0.95 | 0.08 | 0.87 | 18.91 *p* < 0.001 |
| (20) As Time Goes By | 0.14 | 0.74 | 0.60 | −7.44 *p* < 0.001 |
| (21) We Shall Overcome | 0.34 | 0.97 | 0.63 | −11.50 *p* < 0.001 |
| (22) Born to Run | 0.42 | 0.13 | 0.29 | 3.84 *p* = 0.001 |
| (23) Tagesschau | 0.20 | 0.03 | 0.17 | 3.55 *p* < 0.001 |
| (24) Loi Kathong | 0.11 | 0.03 | 0.08 | 1.92 ns |
| (25) Marlboro | 0.67 | 0.97 | 0.30 | −4.13 *p* < 0.001 |
| (26) I'd Like to Teach the World to Sing | 0.19 | 0.97 | 0.81 | −16.26 *p* < 0.001 |

American respondents over 60 had different patterns of recognition than did those under 30. Thus, songs like "Ice Ice Baby," and "Fresh Prince of Bel Air," both popular in the 1990s and early 2000s, were almost universally recognized by the younger age cohort, and not at all by the elderly. The younger group did not recognize though older songs from earlier years, like "As Times go By," "I Love Lucy," and "We Shall Overcome," which were almost universally recognized by over 60s. Ironically, both groups mistakenly recognized the Marlboro commercial intro, as associated with the television show "Bonanza," rather than cigarettes (see Table 7). Recognition of the popular Coca Cola commercial jingle "I'd like to Teach the World to Sing" from the 1970s was spread more evenly between the two groups. "Let it Be" also had a bias toward the older groups. On the other hand, the patriotic song "America the Beautiful" was recognized by both American groups.

**Table 7.** Summary of qualitative comments.

| Song | Summary of Qualitative Comments |
|---|---|
| (1) Let it Be | 159 comments. Is correctly associated with The Beatles and "Let it Be" by most. Low recognition by Thai and Tanzanian. |
| (2) Free Bird | 96 comments. Typically associated with Free Bird and Lynyrd Skynyrd. Recognized primarily in the United States. |
| (3) South African National Anthem | 58 comments. 10 associated with church. 17 Associated it with Africa and/or South Africa. Most positive responses came from Tanzania. |
| (4) America the Beautiful | 144 comments. Most associated it with American patriotism. 9 associated it with Coca-Cola |
| (5) Don't Stop | 92 comments. Associated with Fleetwood Mac. Also "t.v. show." |
| (6) Made in Thailand | 54 comments. Recognized by Thai students, and some of the other students living in Thailand. |
| (7) Ode to Joy | 147 comments. German students recognized it as the European Anthem, Freude Schoene Gottefunken, Americans as "Ode to Joy," and Koreans and Thai as "church music." |
| (8) Somewhere | 242 comments. Associated correctly with Wizard of Oz and/or Judy Garland by about 140. However, some interference from from other singers. But minor. This was recognized properly except in Tanzania. |
| (9) 60 Minutes Clock | 128 comments. Younger Americans recognized it as associated with an American news show. About half associated it with a clock. |
| (10) Moon Represents | 77 comments. Most common response that it was a "Chinese song (13 comments). Four or five wrote the Chinese characters. Few referred to the title in English. Most commonly recognized in Thailand, and by Korean students. |

**Table 7.** *Cont.*

| Song | Summary of Qualitative Comments |
|---|---|
| (11) Chervona Ruta | 24 comments. Four students associated the song with Turkey—one Thai, and three Turkish students. One association with Cambodia, and another with Asia. No students associated it with Ukraine. |
| (12) Somewhere—Iz | 174 comments. Correctly identified in many iterations as the Hawaiian version of Over the Rainbow with a ukulele. Also recognized as part of the Adam Sandler movie "50 First Dates" (5), commercials (5), cartoon music associated with Tom and Jerry (12), and Bruno Mars (2). |
| (13) I Love Lucy | 136 comments. 34 associations with cartoons, particularly Tom and Jerry. The association was strongest among the Korean students. Older Americans tended to associate it correctly with "I Love Lucy" at the highest rates. |
| (14) Happy | 228 comments. Everybody except the older Americans and Tanzanians. Two people thought it was Bruno Mars, but most correctly placed it as Pharrell Williams. |
| (15) Fuer Elise | 228 comments. Most associated with Beethoven and the Fuer Elise. Others included ring tones (7), and piano lessons (34). |
| (16) TID Zeze | 38 comments. Very familiar in Tanzania. Two students in Thailand identified it by name. Nowhere else. |
| (17) Fresh Prince | 102 comments. Mostly from the United States, and younger. |
| (18) Los Tigres del Norte | 69 comments. Students from both Nebraska and Mexico refer to family. Six refer to Germany/Russia. Four refer to polkas. |
| (19) Ice Ice Baby | 167 comments. Most young people associated it with Ice and Ice Baby/Vanilla Ice. Five associated with Queen. Others with MC Hammer. |
| (20) As Time Goes By | 73 comments. Associated with jazz and old movies. |
| (21) We Shall Overcome | 117 comments. 50 associated with church and Christmas. 20 identified it as "We Shall Overcome." |
| (22) Born to Run | 68 comments. 20 identified it correctly. |
| (23) Tagesschau | 114 comments. All Germans recognized it. Others from everywhere except Tanzania identified it as the lead-in to the news. |
| (24) Loi Kathong | 47 comments. Typically associated with Chinese or other Asian music. 6 o7 identify as Thai. No one identified Loi Katong. |
| (25) Marlboro | 140 comments. Primarily associated with Bonanza, and other western shows. No association with Marlboro or commercial. |
| (26) I'd Like to Teach the World to Sing | 60 comments. 15 associated it with Coca-cola or a commercial. Just 7 identified it by name. |

*3.4. Qualitative Comments*

Relevant qualitative summaries of the 26 song intros are presented in Table 7. A selection of observations about the songs is highlighted in the text below.

*3 South Africa/Tanzania National Anthem.* Tanzanians recognized their opening bars of the song that is both Tanzania's national anthem, and that of South Africa, although the lyrics of the song, and the language is different. The Tanzanian recognized the opening words specifically as being the South African version.

*4 America the Beautiful.* This version which was part of a Coca-Cola commercial in 2014. When I first used this in class in 2014, students connected it to Coca-Cola because of its use of Spanish. This was not the case in 2015–2016 when the data were collected for this study—it was widely recognized as simply "America the Beautiful," or other patriotic songs.

*6 Made in Thailand.* by the group Carabao. This iconic 1984 Thai song is about Thai national identity, and the intrusion of foreign products, and is widely played in Thailand. Recognized by most Thai students "of course, I'm Thai!."

*7 Ode To Joy*. Beethoven, vocal version. Recognized everywhere except Tanzania. Americans tended to call it "Ode to Joy" or Christmas music, while the Germans thought of it as the European Anthem. Koreans recognized it as "church music."

*10 The Moon Represents my Heart*. Popular Chinese song by Taiwanese singer Teresa Tang, recorded by her in 1977 and included as a "control." Released in People's Republic of China following the opening up in the early 1980s. Despite the fact it was included as a "control," the song was recognized by Korean and Thai students, but not in Germany, the United States, or Tanzania. This is interesting because this song is known for being one of the most popular songs in the world, but mainly in China and neighboring countries.

*11 Chervona Ruta*. Popular Ukrainian folk song written in 1968. This was included as a control—it is extremely popular in Ukraine. In the countries surveyed, it was recognized mainly by Turkish students who commented that it was "Turkish."

*12 Somewhere/It's a Wonderful Life by the Hawaiian singer Israel Kamakawiwo'ole*. This is a ukelele version of the Wizard of Oz song. This version was recorded in 1990, but did not become popular until after 1999 when it was used in commercials, and movies. Known among all younger respondents, but not among the older Americans. This is in contrast to the original Judy Garland version which was also included in this mix, and was recognized by all, except in Tanzania.

*15 Für Elise*. More Beethoven. Associated with piano lessons, and ring tones. Not recognized only in Tanzania.

*18 Los Tigres del Norte*. "La Mesa del Rincon" Popular "north" Mexican song in the early part of the 21st century. Recognized by my Latino students in California who recalled hearing it because their parents played it frequently. A few German students associated it with classic German polka music.

*19 Ice Ice Baby—Vanilla Ice*. Released in 1990. Everybody except the older Americans and Tanzanians recognized it. A few recognized it as a song from the band Queen.

*21 We Shall Overcome*. US Civil Rights anthem of the 1960s popularized by Martin Luther King. As noted above, this was recognized as the civil rights hymn only by the older Americans. Germans and Koreans recognized it as being gospel and/or church music. Younger Americans did not generally recognize the song.

*23 Tagesschau (Germany)*. Intro to long-running German news program "Tagesschau." All German respondents recognized this, but others recognized as being for the "news," apparently reflecting a musical prelude common in many societies.

*24 Loi Katong (Thailand)*. Popular holiday song in Thailand. This version uses classic Thai notes. This was recognized as "Chinese" by a number of respondents. But not even the Thai students recognized their holiday song. The version used here was in the older Thai pentatonic scale.

*25 Marlboro cigarette Commercial from 1969*. Televised cigarette advertising was banned from American television in 1969. Not even the older Americans associated this intro with cigarettes. Rather they associated it with the television show Bonanza which appeared at the same time.

*26 I'd Like to Teach the World to Sing*. Coca-Cola commercial released in 1971, and re-released with references to the soft drink. Widely recognized in the United States, and occasionally associated with Coca Cola.

## 4. Discussion

### 4.1. The Original Thesis

The question asked at the beginning of this paper had to do with Edward T. Hall's provocative statement:

It can now be said with assurance that individuals are dominated in their behavior by complex hierarchies of interlocking rhythms. Furthermore, these same interlocking rhythms are compatible to fundamental themes in a symphonic score, a keystone in the interpersonal processes between mates, co-workers, and organizations of all types on the interpersonal

> level within as well as across cultural boundaries. I am convinced that it will ultimately
> be proved that almost every facet of human behavior is involved in the rhythmic process.
> (Hall 1983, p. 153)

This is a very reductionist argument, as well as being functional in Durkheim's sense—asserting that musical rhythms underlie *every* or *all* interpersonal processes. This is a large claim against which the data were tested. What the music intro data tell us about this hypothesis is reviewed below. However, it is important to remember that indeed, Hall is writing in very good sociological company, including that of W. E. B. DuBois (1903), and Weber (1978). Most importantly, it is in the tradition of Theodor Adorno (1941) who makes a very similar point about the nature of the popular music for which he held so much disdain. More modern sociological company include most specifically (Tagg 1997), and more generally Frith (1998, 2004), and (Middleton 1990), among others.

But Hall's claim that musical elements (rhythms) are important for ordering society is not only his, and in fact goes back to W. E. B. DuBois who pointed out that rhythm, lyrics, and harmony are what tie together even those living in most oppressive and pre-literate conditions of the world of the American slave. In a different way, Max Weber developed this same theme by noting that the music of Europe was rationalized around Middle C and 12 keys following the invention and commercialization of the piano in early nineteenth century Europe.

This was in turn put in the context of Theodor Adorno's very similar assertion that the capitalist system by the 1940s created music as a commercial product which, via radio, phonograph, and film created images, rhythms, lyrics and sound. As he cynically noted, this is what kept the "common shop girl" working and dreaming in an otherwise exploitative capitalist system. As Adorno noted later in his career, this even extended to the anti-Vietnam War music spread as a commercial product throughout the west in the 1960s. Which brings us back to a much more mundane observation about the popular "ear worms" which as the Australian band "Axis of Awesome" point out in their satirical work "4 Chord Song", spread far and wide, when popular music is reduced to the four chord progression I, V, VI, IV.

### 4.2. What the Data Show

The data described in the results section above show that indeed there are patterns to the recognition of modern commercialized music. In particular, there were strong correlations which could be seen by the Factor Load in the Rotated Components Matrix of Table 1, as well as the Correlations in Tables 2 and 3. These data for example indicate that genres do "travel together," in a manner which is probably patterned both within as well across cultural boundaries as Hall claimed. But it is also true that people in different contexts take different meanings from the same musical elements, e.g., how the Beethoven classic "Ode to Joy" was interpreted as being church music in Asia, the European Anthem in Germany, and "Ode to Joy" in the United States.

The data also show that recognition is patterned by social boundaries which reflect general social, geographic, cultural, and national boundaries. The data reflect a fact that while there is an emerging global musical culture, this culture is not universal. In this sampling, Tanzania was a clear outlier—among other things, there is no clear recognition of Beethoven, just as there is no clear connection by the other cultures of the South African/Tanzanian national anthem. Likewise, it is clear that the American youth and elders also inhabit different musical cultures.

The data also show that music likes at least in the United States, are patterned by generation (See Table 6). Certainly that is the case within the small area of California where the comparison are made. Relevant is the point that music imprints by the age of thirty—which is why something like "As Time Goes By" from the 1942 movie "Casablanca" was so quickly recognized over 70 years after it was first issued.

So as Hall implies, there is a spread of popular rhythms, chord progressions, and other elements of music along cultural lines in a fashion that reflects "interpersonal processes between mates, co-workers, and organizations of all types on the interpersonal level within as well as across cultural boundaries."

There is indeed a global musical culture which stretches across Eurasia and North America just as Adorno feared. This sometimes manifested itself in unexpected ways which were related to cultural and geographic proximity. Popular youth culture in Germany and the United States was similar, though did not extend to other corners of the world.

Tanzania was the big exception in our convenience sample. Students from Tanzania did not recognize rhythms from Beethoven, as well as standards from North American and German culture. On the other hand, students from Korea and Thailand did recognize some—though not all—of the staples of western music. This was the case with the Beethoven, as well as "I Love Lucy," which was recognized as a cartoon-program intro by Korean students.

There were country-specific songs which reflected nationalism and other tendencies, as in the case of "America The Beautiful," "Made in Thailand," and the South African/Tanzanian National Anthem. "Ode to Joy" was perceived as the European song in Germany, but not elsewhere.

Music is also used to brand commercial products. Coca-Cola has been particularly successful in branding with music, though the data indicate that this branding is not necessarily enduring. On the other hand, the American television "Sixty Minutes" did a good job with American youth, although the older Americans missed the association with the television program, perhaps because many were over age thirty when the program first broadcast in 1968 the sound of the ticking clock as a signal to late twentieth century Americans that the weekend was over, and it was time to prepare for the coming week!

The age comparison was only done with American data, but points to an intuitive finding that music tastes are generational, at least since the introduction of commercial radio in the 1930s, which of course was a key variable explaining Adorno's disgust with the "popular music" in the first place.

The correlation of Los Tigres del Norte "La Mesa del Rincon" with "America the Beautiful" presumably reflects the bi-cultural up-bringing of Chicano-Latino students in which both Latino music, and patriotism was recognized. What is surprising perhaps is that Los Tigres did not load with the American pop. Still these respondents who were mostly Chico State students in classrooms which were 40–50% Latino, still loaded most closely with the "American classics" which were also likely to be recognized by the elders from Grass Valley.

What the sum of these observations indicates, is that indeed there is a global popular music culture which stretches from North America to Eurasia, but only weakly to Tanzania. But in all places local music likes do persist, imprinting themselves on the psyche of youth as they pass through adolescence.

### 4.3. Music Recognition as a Methodology in Culture Studies

As E. T. Hall wrote, "individuals are dominated in their behavior by complex hierarchies of interlocking rhythms." The data here do not contradict this, although there is not necessarily evidence for the reductionist form of this generalization. What this study shows is that such hierarchies are culturally bounded as Hall postulated. And among the things bounding group identity are patterns of song recognition that correlate with nationality and generation.[8]

But perhaps the most important finding is that musical intros are a non-intrusive (and fun!) way to assess the inter-personal boundaries that tie individuals together—or not. As Hall, DuBois, Weber, Adorno and others imply, this linkage is largely at the otherwise unknowable sub-conscious level of the isolated romantic but lonely shop girl. Her musical tastes (and economic demands) are presumably imprinted by age thirty. Popular music is a means by which youth struggle with the anomic isolation

---

8    Tagg (Tagg 1997) developed similar ideas about the same time as E. T. Hall, though they did not reference each other. Tagg (Tagg 1997, p. 1) writes of the role of culture and "time sense." He writes of the "cultural skill of decoding the 'meaning' of the sounds in the form of an adequate response." He connects musical elements including tempo, linear time, cyclical times, present time, and hierarchies of duration. This, he claims, is correlated with different traditions from agricultural societies in Europe and Asia. Notably for Tagg this "time sense" is separate from the newer time sense correlated with modern capitalism.

inherent to modern society, and "inter-lock" anonymously with each other as a shared process for recreating social meaning.

Confirmed also is the persistence of melodies across what for the older people was a lifetime. Some of these rhythms are of course passed onto children (hats off to Beethoven and patriotism), while others are not (sorry Bruce Springsteen, Beatles, and We Shall Overcome!).

Finally, the data highlight the fact that the elements of music, and therefore music recognition is bounded, particularly by age, and culture. Linkages between cultures can though be surprising, such as the recognition of the north Mexican song by Germans (but not Americans), and the spill into Turkey of the Ukrainian song Chervona Ruta. The song "The Moon Represents my Heart" moved into Thailand (and to a lesser extent Korea), from China/Taiwan.

### 4.4. Future Studies

This paper focuses on music recognition across nations, cultures, and generations. And while a hypothesis about the transmission of musical elements and rhythm is embedded in it, this is primarily a paper describing a new methodology for measuring cultural chasms of sub-conscious music reception, whether geographical, national identity, or by age cohort. It is a method for assessing the recognition of how musical elements vary between populations. The study design focused on cross-national variation, and a bit on generational differences in music likes. But it could also have focused on the transmission of rhythms within the United States by ethnic group. In California, most relevant would have been comparing recognition of the rhythms of "Let it Be", Lost Tigres del Norte's "La Mesa del Rincon," and "We Shall Overcome," which probably pattern by ethnicity, as well as by age. The same may well be the case if data from elders were collected in Thailand, Korea, Tanzania, or elsewhere.

For example, a provocative experiment in Thailand would be to assess differences between recognition of songs on a pentatonic scale found in traditional Thai music, and the global 12-note chromatic scale of Weber's piano. Our impression is that over the last thirty or forty years, there has been a shift between the local rhythms, and the more globalized western music rooted in the 12-note scale now common in Thai popular music. Thus, a popular Thai holiday song like "Loi Katong" is played in both scales, with perhaps a higher recognition of the pentatonic version by older people, and a lower recognition by younger. This may explain why the first bars of Loi Katong (song 26) were not recognized by the Thai students. Perhaps it would have been recognized by older Thai more readily.

Ultimately, though this paper is describing a robust methodology that can be used to unobtrusively measure what Durkheim called social cohesion. Cohesive groups are likely to have similar patterns of recognition rooted in the shared musical experiences of youth. In this sense, the technique is a least as good as a more traditional Likert scale questionnaire to evaluate what both Adorno and Weber implicitly assert is not only the rationalization of music, but more generally of culture itself.

Political groupings may also be measurable using this technique. Music likes are closely tied to emotions, as are political loyalties. Indeed, the close ties between political loyalty is why governments routinely promote patriotic songs in schools. It has not escaped our attention that this technique might be used to evaluate the distances between cultural relationships between groups seeking to reconcile after war, and other conditions where political unity is tested. A country which is tied together through the similar rhythms of "ear worms" is more likely to work together well in a military, school, factory, office setting, and even government as E. T. Hall (1983) implied in his book *The Dance of Life*, and Adorno (1941) did in his descriptions of musical genre. This is perhaps why East and West German reunification of populations embedded in Beethoven somehow worked better than say, the American occupation of Iraq where there was little shared between, the Egyptian music of Um Kulthum who is extraordinarily popular in the Arab world, and Fresh Prince of Bel Air which was the music tying American soldiers to each other. This ultimately is the main point made by theorists like Hall, Weber, DuBois, and Adorno about musical genre. And again while Hall's extreme assertion may not be confirmed by this research, there is certainly nothing to disconfirm it, either. The study of the

transmission of musical elements and how they reflect the organization of social groups is a field open for exploration.

## 5. Conclusions

This paper explored empirically Hall's question about the role of musical elements, including what Hall called "rhythm" in ordering group identity. There was empirical backing for this assumption, though we would not conclude with Hall's more reductionist conclusions. A more important conclusion is that this is an effective method for describing the elements of cultural relatedness.

**Author Contributions:** Conceptualization was done by T.W. and D.P. Theoretical development was primarily by T.W. while D.P. was primarily responsible for the experimental design and the statistical analysis. Drafting of the article was by T.W. T.W. and D.P. both edited.

**Funding:** This research received financial support from the professional development funds from the Department of Sociology, California State University, Chico.

**Acknowledgments:** Marcos Zepeda helped with selecting the song intros included in this paper in 2014–2015. Jamie Simpson from Midland College assisted with administering the questionnaires in Nebraska, and delivering them to Tanzania. The Rev. John Materu administered questionnaires in Tanzania. Christina L. Quigley provided editorial advice about musicology. Students and faculty at Leuphana University in Germany who informed ideas for this article include Guenter Burkart, Volker Kirchberg, Lynette Kirschner, Yvonne Foerster, Lukas Iden, Benjamin Elbers as well as others. In Thailand, the paper was enriched through discussions with Wutthichula Khunpatwattana, Eva Mazharenko, and students from the General Education program of the International College.

**Conflicts of Interest:** The authors declare no conflict of interest.

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
