# Peer review of "Cross-National Attunement to Popular Songs across Time and Place: A Sociology of Popular Music in the United States, Germany, Thailand, and Tanzania"

_socsci, doi:10.3390/socsci8110305_

Round 1

Reviewer 1 Report

This is a potentially interesting paper that describes something about musical recognition across different contexts. Unfortunately, the theoretical grounding of the paper is unsound (there is no longer any need to empirically argue with Adorno) and the engagement with the wider literature is lacking (the authors seem unaware that broadly similar questions have been studied and debated for decades in popular music studies and music sociology). I was also unclear about the overall argument of the essay. If, as the authors say, the main contribution of the essay is to present a new methodology, then perhaps this should be the focus in future revisions.

I am not commenting on the quantitative research design or execution, as I am not qualified to do so.

Author Response

References to a selection of the popular music, and sociology of music has been added tp the article., including especially those referring to Frith and Middleton.  In doing this review, we came across Tagg's paper, and I do think that this adds a great deal to the argument. This is done in order to tie the article more carefully to the traditions of the Sociology of Popular Culture,and musicology, especially the understandings that come out of the traditions of the Birmingham School.  This is why reference was also made to Storey.

The reviewer suggests that we emphasize that this article proposes a methodology, which currently was only mentioned in the Discussion section.  To emphasize that this is indeed an article proposing a method for understanding the "travel" of musical elements, we have added a line to the abstract, and in the Introduction.  This matches the emphasis on this conclusion described in the discussion section.

We too like Adorno very much, but we do think that this article adds an empirical basis for Adorno's argument. Adorno's older article (1941) continues to intrigue students, and as we wrote this argument fits in well with even older understandings of music inherited from DuBois and Weber.  Accordingly we made more systematic reference to the literature which addresses how Weber in particular viewed the rhythms of society and modernity.

Reviewer 2 Report

My overall evaluation is that this article represents an original and interesting work that presents new knowledge on listening habits to popular songs. The article is original on the topic of cross-national attunement to popular songs across time and place. In this way, this article is highly interesting with an original contribution to the research on how – and to what extent – popular songs enters the collective memory over time. What is interesting is the comparison of listening habits across national boundaries and age groups in The United States, Germany, Thailand, and Tanzania. This is carried out over time using quantitative survey research and qualitative thought provoking and interesting assessments and discussions. In that sense, the article works well on its own terms. But I do have some suggestions for improvement that the authors should consider: The topic of the article is relevant both in popular music studies (including popular musicology) and media studies. Accordingly, the article will be strengthened if it incorporates a brief discussion on how common, or not, this type of study is within popular music research and / or media studies. I miss references to works by key popular music scholars such as Richard Middleton, Simon Frith, Stan Hawkins, Allan F. Moore, Peter Wicke and others who all contributed significantly in the study of popular songs and listening habits. In fact, almost the entire popular musicology field is more or less concerned on listening habits within popular music, how the brain perceives songs at the moment, etc. Here I recommend the author conducts searches for previous research in the field. This does not have to be a comprehensive point in the article, but the research field today across several disciplines should at least be mentioned and where reference is made to key research work. The standard of scholarship presented (viz. findings, methodology, interpretations) is acceptable. The article is well structured, and subject matter, theories, methodologies, findings and interpretations are coherently and clearly formulated and presented. Finally, a small detail, author names referenced should be checked for spelling. The most notable is Theodor W. Adorno, which is consistently written Teodor Adorno in the text. I recommend the article for publication after revision.

Author Response

We have added references to Frith and Middleton in particular, at the suggestion of this reviewer.  In addition, while reading these references, I found the article by Tagg, which emphasizes the role of timing in music, and is directly relevant to this article. In to process of doing this. Finally, we also added some references to the Birmingham School of Cultural Sociology (i.e. Storey's book about Sociology of Culture and Stuart Hall).  we also looked at the papers by Hawkins, Wicke, and Moore.

The references recommended by Reviewer 2 are very interesting, and relevant to understandings of what our paper is trying to say. In reviewing the recommended references, we were impressed with their relevance to the more general theme highlighted in our paper.  Making reference to them, albeit in at times in a perfunctory was very helpful, and we appreciate being pushed in that direction.     In a future paper, we would like to assess more systematically how the literature on popular music and musicology can be used to understand cross-cutural recognition of music patterns.

Round 2

Reviewer 1 Report

I have included comments to the journal editors